# Enhance ZnO Photocatalytic Performance via Radiation Modified g-C_3_N_4_

**DOI:** 10.3390/molecules27238476

**Published:** 2022-12-02

**Authors:** Yayang Wang, Xiaojie Yang, Jiahui Lou, Yaqiong Huang, Jian Peng, Yuesheng Li, Yi Liu

**Affiliations:** 1School of Chemistry and Chemical Engineering, Wuhan University of Science and Technology, Wuhan 430074, China; 2School of Nuclear Technology and Chemistry & Biology, Hubei Key Laboratory of Radiation Chemistry and Functional Materials, Hubei University of Science and Technology, Xianning 437000, China; 3Institute for Superconducting and Electronic Materials, Australian Institute for Innovative Materials, Innovation Campus, University of Wollongong, Squires Way, North Wollongong, NSW 2522, Australia; 4College of Chemistry and Chemical Engineering, Tiangong University, Tianjin 300387, China

**Keywords:** g-C_3_N_4_, g-C_3_N_4_/ZnO, electron beam irradiation, photocatalytic performance

## Abstract

Environmental pollution, especially water pollution, is becoming increasingly serious. Organic dyes are one type of the harmful pollutants that pollute groundwater and destroy ecosystems. In this work, a series of graphitic carbon nitride (g-C_3_N_4_)/ZnO photocatalysts were facilely synthesized through a grinding method using ZnO nanoparticles and g-C_3_N_4_ as the starting materials. According to the results, the photocatalytic performance of 10 wt.% CN-200/Z-500 (CN-200, which g-C_3_N_4_ was 200 kGy, referred to the irradiation metering. Z-500, which ZnO was 500 °C, referred to the calcination temperature) with the CN-200 exposed to electron beam radiation was better than those of either Z-500 or CN-200 alone. This material displayed a 98.9% degradation rate of MB (20 mg/L) in 120 min. The improvement of the photocatalytic performance of the 10 wt.% CN-200/Z-500 composite material was caused by the improvement of the separation efficiency of photoinduced electron–hole pairs, which was, in turn, due to the formation of heterojunctions between CN-200 and Z-500 interfaces. Thus, this study proposes the application of electron-beam irradiation technology for the modification of photocatalytic materials and the improvement of photocatalytic performance.

## 1. Introduction

With increasing industrialization, environmental pollution is becoming more and more serious, especially water pollution [1,2,3]. Therefore, the treatment of industrial wastewater from various sources has attracted extensive attention from researchers. It is an important challenge to remove pollutants from bodies of water. Among the various types, organic dyes are some of the most harmful pollutants that pollute groundwater and destroy ecosystems. Photocatalysis can effectively degrade organic dyes and is widely used in the degradation of these types of pollutants [4,5].

Photocatalytic technology, due to the excellent stability of the catalyst, its low cost, and its environmental friendliness, has been widely studied by researchers [6,7,8]. ZnO, with good chemical stability, has low toxicity, low cost, and many other advantages, thus, it has been extensively studied in the field of photocatalytic degradation [9,10,11,12]. The narrow spectral absorption (Eg = 3.37 eV) and low degree of separation of photogenerated electron–hole pairs of ZnO, however, limit its applications in photocatalysis [13]. Improving its light absorption capacity and promoting the separation of photogenerated electron–hole pairs can effectively improve the photocatalytic performance of ZnO [14,15,16,17]. Graphitic carbon nitride (g-C_3_N_4_) possesses many good characteristics, such as a low band gap (Eg = 2.7 eV), stable physical and chemical properties, good absorption capacity of visible light, simple synthetic methods, thermal stability, and low cost [18,19,20,21,22]. To further improve the performance of g-C_3_N_4_, electron beam radiation technology is used [23]. Enhancement of the photocatalytic activity of ZnO can be achieved by compounding g-C_3_N_4_ with it [24,25,26,27,28]. Ding et al. [29] reported a composite catalyst, AgCl/ZnO/g-C_3_N_4_, which was prepared via calcination, hydrothermal reaction, and in-situ deposition processes. The catalyst was assessed for its photocatalytic efficiency in eliminating tetracycline hydrochloride from pharmaceutical wastewater under visible light. Ganesh et al. [30] prepared ZnO, Sb-doped ZnO, and ZnO: Sb/g-C_3_N_4_ nanocomposite using a simple chemical route. ZnO: Sb and g-C_3_N_4_, which reduced luminous intensity due to increased charge transfer, can effectively improve photocatalytic performance. The degradation efficiency of the ZnO: Sb/g-C_3_N_4_ sample was 86% in 60 min. The efficiency of the degradation of methylene blue (MB) (1 × 10^−5^ M) by ZnO: Sb/g-C_3_N_4_ was 86%. Based on these studies, we modified the materials by electron beam radiation to study their degradation efficiency against dyes in high concentrations.

In this study, ZnO was prepared at different annealing temperatures and its photocatalytic properties were examined. The carbon nitride photocatalysts, g-C_3_N_4_ (CN-X (where X is the radiation absorption dose and X = 100, 200, 300, and 400 kGy)), were prepared by direct calcination of urea and CN-200 was prepared by the electron beam radiation technique. The presence of the CN-200/Z-500 heterostructures produced a large number of catalytic active sites, enhancing the catalytic performance. It was found that 10 wt.% CN-200/Z-500 particles degraded 98.9% of MB (20 mg/L) in 120 min. The improvement of the photocatalytic performance of 10 wt.% CN-200/Z-500 nanocomposite resulted from the improvement of the separation efficiency of photoinduced electron–hole pairs, which was due to the formation of heterojunctions between CN-200 and ZnO interfaces.

## 2. Results and Discussion

### 2.1. XRD and FT-IR Analysis

The crystal phases of the samples were examined by XRD. Figure 1a–d shows the XRD patterns of g-C_3_N_4_, ZnO, and CN-200/Z-500. g-C_3_N_4_ has its strongest peak at 2θ = 27.33° corresponding to the (002) planes (Figure 1a) due to the typical interplanar stacking structures of graphitic materials. The g-C_3_N_4_, which was irradiated by an electron beam, shows a red shift for the reflection of the (002) crystal planes. Electron beam irradiation was beneficial in increasing the spacing between crystal planes. To investigate the influence of different annealing temperatures on the crystalline properties of ZnO, the prepared materials were investigated by XRD, as presented in Figure 1b. Obviously, the peak position did not change with annealing temperature, suggesting that the crystal form did not change after calcination due to the material temperature. The crystal forms of CN-200, Z-500, and 10 wt.% CN-200/Z-500 are shown in Figure 1c. The diffraction peak of CN-200 at 27.65° corresponds to the (002) planes (JCPDS NO. 87-1526). The high-intensity diffraction peak of CN-200 at 27.65° corresponds to the packing of the conjugated aromatic system. Z-500 exhibits prominent characteristic peaks corresponding to the hexagonal wurtzite phase at 31.77°, 34.42°, 36.25°, 47.54°, 56.60°, 62.86°, 66.38°, 67.96°, 69.10°, 72.56°, and 76.95°, corresponding to the (100), (002), (101), (102), (110), (103), (200), (112), (201), (004), and (202) planes (JCPDS NO. 36–1451), respectively. The diffraction peaks at different positions are well-matched with the wurtzite diffraction peaks. This showed that the prepared hexagonal wurtzite zinc oxide samples have high crystallinity. Figure 1d presents the FT-IR spectra of CN, CN-200, Z-500, and 10 wt.% CN-200/Z-500. The peak at 815 cm^−1^ in the spectrum of CN corresponds to the s-triazine ring system. The peaks at 1242.35 cm^−1^ and 1627.17 cm^−1^ correspond to the C–N and C=N stretching vibrations, respectively. Comparing the infrared spectra of CN and CN-200, it can be seen that the electron beam irradiation did not change the structure of graphitic carbon nitride. In the case of Z-500, the band at 438.32 cm^−1^ due to the Zn–O stretching mode indicates the formation of ZnO crystal. The FT-IR spectrum of 10 wt.% CN-200/Z-500 was similar to the characteristic spectra of both CN-200 and Z-500, suggesting the retention of the typical graphitic structure of CN-200 after it was compounded with Z-500.

### 2.2. XPS Analysis

X-ray photoelectron spectroscopy (XPS) was used to analyze the chemical elements on the surface of the composite material. The XPS spectra of C 1s, N 1s, O 1s, and Zn 2p are shown in Figure 2. Three peaks were observed, which were at 288.54 eV, 287.13 eV, and 285.15 eV, respectively, corresponding to the C=N, C–N, and C–C bonds in Figure 2a. The peak at 287.13 eV belongs to the N–C=N bonds with sp^2^ hybridization. The peak at 288.54 eV was assigned to sp2-bonded C–NH_2_. As shown in Figure 2b, the N 1s spectrum of 10 wt.% CN-200/Z-500 has three characteristic peaks with binding energies of 401.19, 399.09, and 398.87 eV. The peak at 401.19 eV was attributed to C–N–H bonds, while the other peaks located at 399.09 and 398.87 eV were ascribed to N–(C)3 and C=N–H, respectively. The XPS spectrum of Zn 2p exhibits two peaks at 1045.17 and 1022.05 eV (Figure 2c), which correspond to Zn 2p_1/2_ and Zn 2p_3/2_, respectively. The Zn 2p peaks of 10 wt.% CN-200/Z-500 showed a negative shift in comparison with the pristine ZnO, which was reported by the authors of [31,32]. As shown in Figure 2d, three peaks were resolved from the O 1s spectrum of 10 wt.% CN-200/Z-500. The peak with binding energy of 530.50 eV was determined to belong to the lattice oxygen of Zn–O, which further verified the successful combination of Z-500 and CN-200. The other two peaks, located at 532.55 and 532.45 eV, were assigned to water molecules and surface hydroxyl, respectively.

### 2.3. FE-SEM and HR-TEM Analysis

The FE-SEM images of the CN, CN-200, Z-500, and CN-200/Z-500 samples are shown in Figure 3. A typical multilayer stacking structure of the CN is shown in Figure 3a,b. Compared with CN, CN-200 shows better dispersion, as shown in Figure 3c,d. Figure 3e,f indicates that Z-500 was mainly composed of non-uniform spherical particles. It is clear from Figure 3g,h that the composite material CN-200/Z-500 was formed from nanosheets of CN-200 and nanoparticles of Z-500. Figure 3g,h presents an SEM image of 10 wt.% CN-200/Z-500 photocatalyst. As seen above, the Z-500 photocatalyst was composed of irregular nanoparticles, which contributed to its low photocatalytic activity under visible light. The SEM image of the 10 wt.% CN-200/Z-500 shows that the Z-500 was evenly distributed on the surface of the CN-200, which favored the formation of heterojunctions and resulted in enhanced photocatalytic activity.

The 10 wt.% CN-200/Z-500 with the best photocatalytic activity was further analyzed by TEM and high-resolution TEM (HR-TEM). Figure 4a,b revealed typical TEM images of the composite photocatalyst 10 wt.% CN-200/Z-500. Z-500 dispersed on the surface of CN-200 can be seen in Figure 3 and Figure 4. This observation is consistent with the XRD and SEM results.

### 2.4. Photocatalytic Activity

The photoluminescence (PL) emission spectra was conducted to determine the recombination rate of photoinduced charge carriers. The properties of photogenerated charge carriers between CN-200, Z-500, and 10 wt.% CN-200/Z-500 were studied using 365 nm as the excitation wavelength. As shown in Figure 5a, the PL strength of 10 wt.% CN-200/Z-500 was much weaker than that of pure CN-200, indicating that the photoinduced charge carrier recombination rate of 10 wt.% CN-200/Z-500 composite is greatly reduced. Figure 5b showed UV–visible spectra of 10 wt.% CN-200/Z-500 composites, CN-200 and Z-500. The corresponding band gaps of ZnO and g-C_3_N_4_ were acquired from the indirect estimation of Tauc plot (Ahv)^n^ vs. The estimated Eg values for Z-500, CN-200, and 10 wt.% CN-200/Z-500 composites were 3.24 eV, 2.89 eV, and 3.22 eV, respectively. Z-500 had a narrower band gap after doping with CN-200, which indicated that it can absorb more visible light and show a better light response under visible light irradiation.

### 2.5. Photocatalytic Activity

Figure 6a,b shows the photocatalytic activities of different catalysts for MB (20 mg/L) degradation. The catalytic performance of ZnO prepared at different calcination temperatures while keeping the amount of CN-200 doping constant is shown in Figure 6a. The results showed that 10 wt.% CN-200/Z-500 exhibited the best catalytic activity. A very rapid degradation rate of MB was observed initially and then it slowed down until no more degradation occurred. As can be seen from Figure 6a, Z-500 showed better MB degradation than the other Z-T samples (T = 400, 600, and 700 °C). It is noteworthy that Z-500 and CN-200 played a crucial role in the construction of effective heterojunctions between the two components. As expected, the photocatalytic activity of CN-200/Z-500 composite was enhanced as compared to Z-500. Under visible light irradiation, the 10 wt.% CN-200/Z-500 decomposed MB solution (20 mg/L) with a high degradation rate of 98.9% within 120 min. This showed that CN-200 was well incorporated into Z-500, effectively enhancing the photocatalytic performance of the composite under visible light irradiation. Figure 6b shows the influence of different composite proportions on photocatalytic performance. It can be seen that, when the doping ratio was 10 wt.%, the photocatalytic performance was the best.

MB dye was used to simulate pollutants and the photocatalytic activities of the catalysts were assessed by studying the MB degradation efficiency under visible light. The degradation of MB was monitored by observing the decrease in the intensity of the characteristic absorption peak at 662 nm with time. The degradation reaction kinetics of MB can be expressed quantitatively with the use of the pseudo-first-order model: ln(C_0_/C_t_) = kK_t_ = K′_t_, where C_0_ refers to the original dye concentration; C_t_ refers to the MB concentration; K′ represents the pseudo-first-order rate constant; and t represents the irradiation time. In accordance with this equation, K′ can be acquired from a linear plot of ln(C_0_/C_t_) against t, which represents the degradation rate and is proportional to the photocatalytic degradation rate.

The kinetics of photo-oxidation/photoreduction of MB over the different samples are presented in Figure 6c,d. The MB photo-oxidation/photoreduction kinetics of different samples are shown in Figure 6c,d and K′ is obtained after data fitting. Specific data are shown in Table 1 and Table 2. Combined with the data analysis, the photocatalytic degradation of MB conforms to the first-order kinetic model, R^2^ > 0.92, indicating a good degree of fit. Based on Figure 6d, the photocatalytic degradation efficiency is 98.86% for the 10 wt.% CN-200/Z-500, 80.97% for the Z-500, and 89.64% for the 5 wt.% CN-200/Z-500 after 120 min.

The cyclic stability of 10 wt.% CN-200/Z-500 composite catalyst was evaluated by cyclic experiments of photocatalytic degradation of MB. The used catalyst would be centrifuged at high speed after each cycle test, washed, and vacuum-dried overnight at 60 °C for collection before use. Figure 7a showed the photocatalytic efficiency curves of 10 wt.% CN-200/Z-500 in three cycles. As shown in Figure 7a, after three cycles, the degradation rate of MB was 91.7%, while the degradation rate of the first cycle was 98.4%, indicating that the degradation efficiency of the catalyst did not decrease significantly with the progress of the photocatalytic process. The results indicated that the prepared 10 wt.% CN-200/Z-500 catalyst was stable and reusable in the photocatalytic degradation process. The trapping experiments were carried out to explore the photocatalytic mechanism. To test the role of these reactive species, Ethylenediaminetetraacetic acid disodium salt (EDTA-2Na), isopropyl alcohol (IPA), and β-benzoquinone (BQ) were employed as scavengers for h^+^, •OH, and •O_2_^−^, respectively. As can be seen in Figure 7b, the degradation ratios for MB were all reduced after the addition of the three trapping reagents, indicating that the h^+^, •OH, and •O_2_^−^ are all responsible for the degradation process. When the EDTA-2Na was added into the reaction system, the degradation efficiency is remarkably decreased, suggesting that h^+^ takes a crucial part in the degradation of organic pollutants.

On the basis of the results of the photodegradation and photogenerated carrier trapping test, a possible photodegradation mechanism for 10 wt.% CN-200/Z-500 was explained to clarify the enhanced photocatalytic activity of it for the degradation of MB. The degradation mechanism was shown in Figure 8. CN-200 absorbs visible light to induce Π–Π* transition, transporting the excited-state electrons from the HOMO to the lowest unoccupied molecular orbital (LUMO). The LUMO potential of CN-200 (−1.12 eV) is more negative than the conduction band (CB) edge of Z-500 (−0.5 eV), so the excited electron on CN-200 could directly inject into the CB of Z-500. Meanwhile, the holes on the VB of Z-500 transfer to the CN-200 because the VB edge potential of Z-500 was more positive than the CN-200. This can effectively restrain the recombination of photoinduced electron–hole pairs for Z-500 and CN-200. The O_2_ molecules trapped photoelectron on the surface of 10 wt.% CN-200/Z-500 and generate the superoxide anion radical (•O_2_^−^). Due to its high activity and disability of penetrating in bulk of the sample, •O_2_^−^ mostly degrades the adsorbed MB molecules on the surface of the catalyst. Simultaneously, h^+^ reacted with water molecules to produce hydroxyl radicals. The resulting •OH and h^+^ oxidize the organic pollutants into small molecules, such as CO_2_ and H_2_O [33]. The photocatalytic reaction process is described below:CN-200 + hv → h^+^ + e^−^
Z-500 + hv → h^+^ + e^−^
CN-200 (e^−^_CB_) → Z-500 (e^−^_CB_)
Z-500 (h^+^_VB_) → CN-200 (h^+^_VB_)
Z-500 (e^−^) + O_2_ → •O_2_^−^
•O_2_^−^ + 2H^+^ + e^−^ → H_2_O_2_
H_2_O_2_ + e^−^ → •OH + OH^−^
h^+^ + •OH + •O_2_^−^ + pollutants → CO_2_ + H_2_O + degradation intermediate

## 3. Experimental

### 3.1. Preparation of CN-X by Irradiation (X = 100, 200, 300, and 400 kGy)

g-C_3_N_4_ preparation: 20 g urea (CH_4_N_2_O) was calcined at 550 °C with a heating rate of 5 °C·min^−1^ for 180 min in an alumina crucible in a muffle furnace. The sample was crushed into powder after calcination.

Sample preparation for irradiation: 0.12 g g-C_3_N_4_, 12 mL isopropanol, 120 μL water, and 240 μL ammonia were mixed in a reagent bottle. The mixture was ultrasonically stirred for 10 min, which was repeated three times. The sample was transferred to a polyethylene bag and sealed under a vacuum. The sample was irradiated by a 1 MeV electron accelerator (1 MeV; Wasik Associates, Dracut, MA, USA) with a dose rate of 20 kGy/pass. The sample was repeatedly washed with ethanol and water, subsequently dried in an oven at 60 °C, and ground for later use. The samples with different irradiation doses of CN were denoted as CN-X, where X is the radiation absorption dose and X = 100, 200, 300, and 400 kGy, respectively.

### 3.2. Preparation of ZnO

ZnO was prepared using the co-precipitation method. Solution A was prepared by dissolving ZnSO_4_ (100 mmol) and cetyltrimethylammonium bromide (CTAB, 4.3 mmol) in 125 mL of distilled water. Solution B consisted of ammonium bicarbonate (50 mmol) dissolved in 125 mL of distilled water. Solution B was then quickly poured into solution A, sonicated, and stirred. A gray precipitate resulted, which was then stirred for 60 min. The sample was washed several times with ethanol and water and then dried at 80 °C. Finally, it was calcined for 3 h at the rate of 2 °C·min^−1^ at 400, 500, 600, and 700 °C, respectively. The samples were labeled as Z-T (T = 400, 500, 600, and 700 °C).

### 3.3. Synthesis of CN-200/Z-500

During the earlier studies, it was found that the best photocatalytic efficiency for C_3_N_4_/ZnO was obtained when the annealing temperature was 500 °C and the radiation absorption dose was 200 kGy. Therefore, CN-200 and Z-500 were compounded together by the grinding method. CN-200 (50 mg) was added to the desired mass of Z-500. During the grinding process, 5 mL of absolute ethanol was added every 5 min, and after grinding for 30 min, the mixture was transferred to an oven at 60 °C and dried for 24 h.

### 3.4. Characterization

The crystalline samples were characterized by an X-ray diffractometer (XRD, DMAX-D8X; Rigaku, Tokyo, Japan). The dye degradation was studied by using an ultraviolet-visible (UV-Vis) (TU-1950; Persee, suzhou, China). The morphologies of the materials were analyzed by field-emission scanning electron microscopy (FE-SEM, SU8220; Tokyo, Japan) and field emission transmission electron microscopy (FE-TEM, FEI Tecnai G2 F30; Hillsboro, OR, USA). KBr pellets of the samples were made to analyze the samples using Fourier-transform infrared spectroscopy (FT-IR) (Thermo Ferret iS10 infrared spectrometer, Thermo Fisher Scientific, Waltham, MA, USA). Photoluminescence spectra (PL) of samples was recorded on an Edinburgh FLS1000 fluorescent spectrophotometer (UK). Ultraviolet–visible diffuse reflectance spectra (UV-Vis DRS; Japan) was performed on a Shimadzu, UV-3600i Plus UV-Vis spectrometer with a scan range of 200–800 nm.

### 3.5. Photocatalytic Experiments

The photocatalytic activities of the samples were evaluated by using the dye MB as the model. All experiments were performed in an open system at room temperature. The photocatalyst (50 mg) was dispersed in 50 mL of MB solution (20 mg/L) and stirred for 30 min in dark circumstances to achieve adsorption–desorption equilibrium between the photocatalyst and the MB dye in the photocatalysis study.

The catalysis reaction was performed under visible light by taking out 2 mL of solution from each sample suspension every 20 min. High-speed centrifugation was employed to remove photocatalysts. The concentrations of MB at various reaction times were measured with a UV-Vis spectrophotometer at λmax = 662 nm, and the degradation rate (%) was calculated using the following equation:(1)$$\mathrm{Degradation}\;\mathrm{Rate}\;\left(\%\right)=\frac{C_{0}-C_{t}}{C_{t}}\times 100\%=\frac{A_{0}-A_{t}}{A_{0}}\times 100\%$$

Here, *C_t_* and *C*_0_ represent pollutant concentrations at times *t* and *t*_0_, respectively. *A*_0_ and *A_t_* are the absorbances of the MB solution before irradiation and after irradiation, respectively.

## 4. Conclusions

In this study, ZnO was prepared at different annealing temperatures and its photocatalytic properties were examined. The best photocatalytic degradation of 20 mg/L MB dye was achieved by ZnO prepared at 500 °C under visible light radiation. g-C_3_N_4_ photocatalysts were prepared by direct calcination of urea and CN-200 was prepared by irradiation with electron beams. The radiation dose of CN-200 was 200 kGy. A series of CN-200 doped ZnO hetero-structured photocatalysts were successfully prepared using the grinding method. The samples were characterized using XRD, FT-IR, FE-SEM, and HR-TEM. MB was chosen as the photodegradation target, and the photocatalytic activity of the composites was evaluated. Interestingly, 10 wt.% CN-200/Z-500 particles degraded 98.9% of the MB in 120 min, owing to the presence of a large number of catalytic active sites in CN-200/Z-500 heterostructures that enhanced its catalytic performance. Thus, the 10 wt.% CN-200/Z-500 has great potential to degrade pollutants. Thus, it can be concluded that electron beam irradiation technology can be effectively used for the modification of photocatalytic materials and the improvement of their photocatalytic performance.

## Figures and Tables

**Figure 1 molecules-27-08476-f001:**
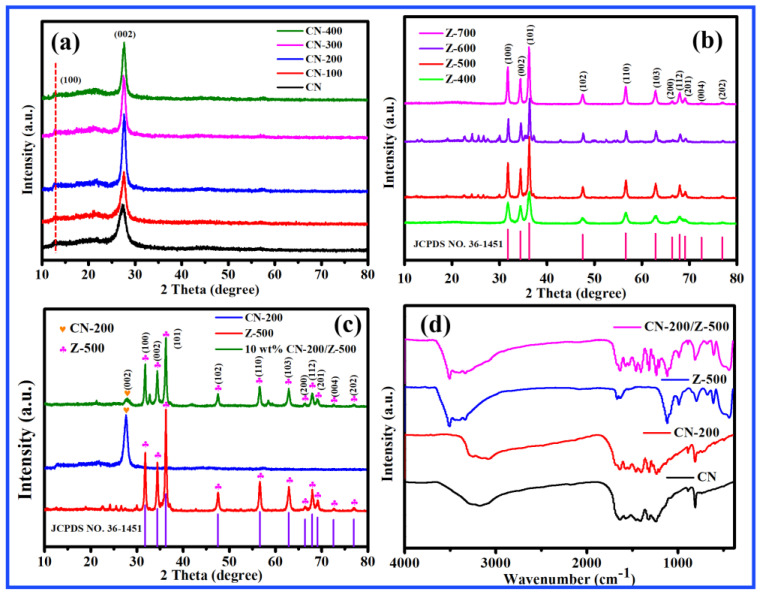
(**a**) XRD patterns of CN prepared under different absorbed radiation doses; (**b**) XRD patterns of ZnO prepared at different calcination temperatures; (**c**) XRD patterns of CN-200, Z-500, and 10 wt.% CN-200/Z-500; (**d**) infrared characterization of CN, CN-200, Z-500, and 10 wt.% CN-200/Z-500.

**Figure 2 molecules-27-08476-f002:**
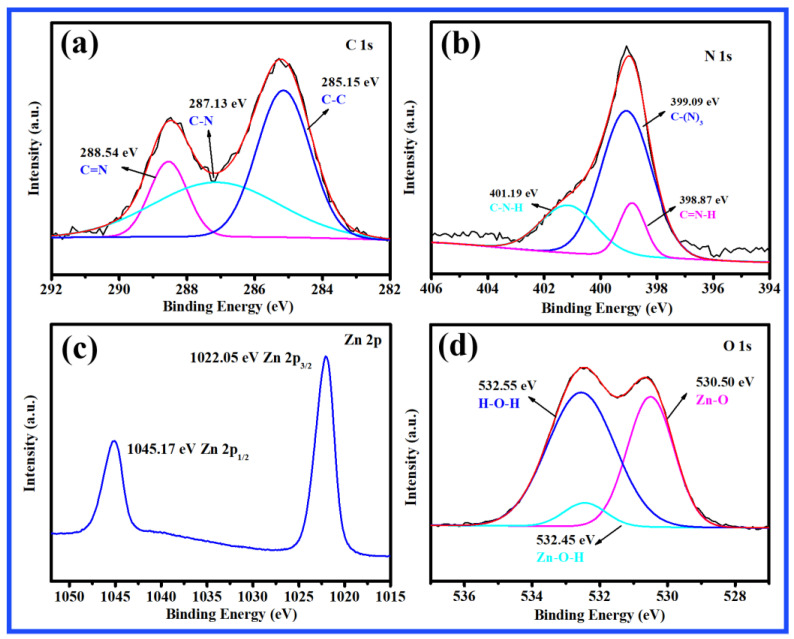
XPS spectra of 10 wt.% CN-200/Z-500: (**a**) C 1s; (**b**) N 1s; (**c**) Zn 2p; (**d**) O 1s.

**Figure 3 molecules-27-08476-f003:**
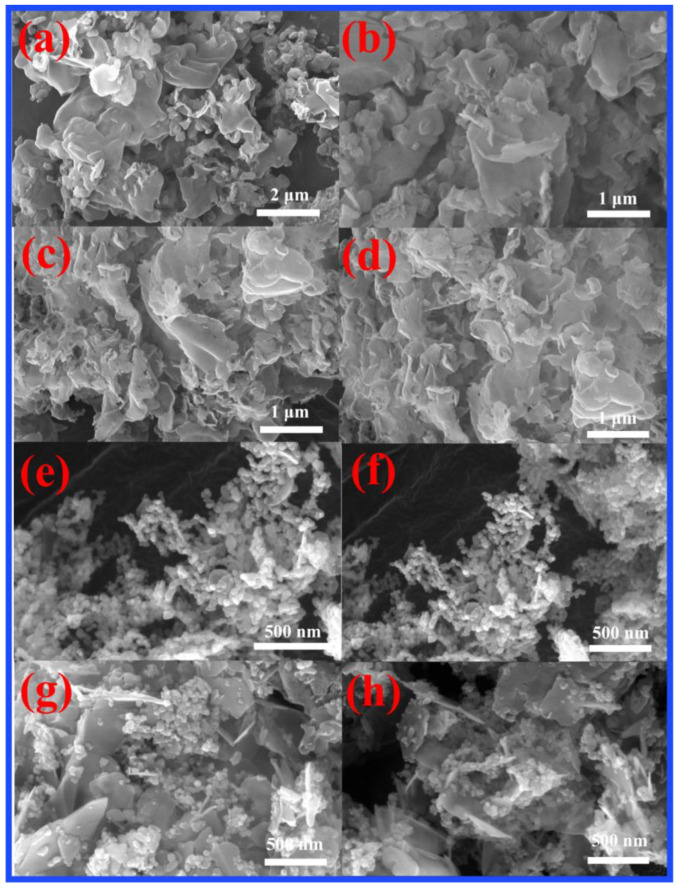
(**a**,**b**) SEM images of CN; (**c**,**d**) SEM images of CN-200; (**e**,**f**) SEM images of Z-500; (**g**,**h**) SEM images of CN-200/Z-500 photocatalyst.

**Figure 4 molecules-27-08476-f004:**
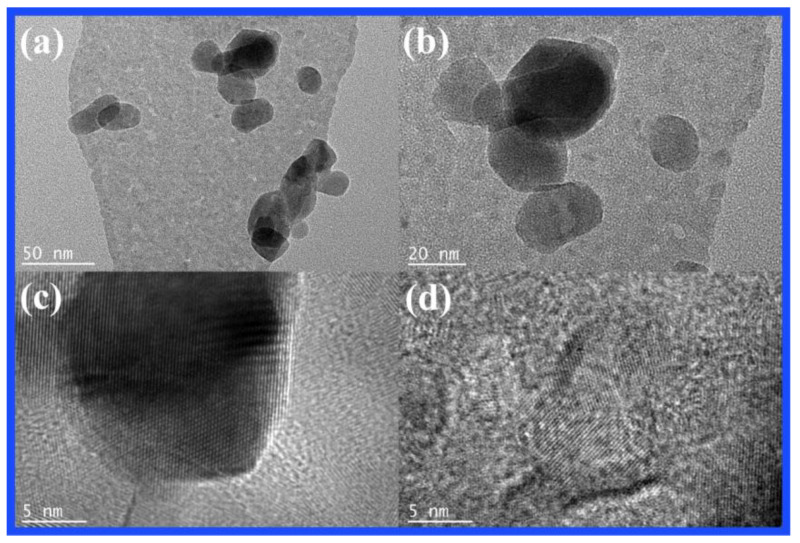
(**a**,**b**) TEM images of CN-200/Z-500 photocatalyst; (**c**,**d**) HR-TEM images of CN-200/Z-500 photocatalyst.

**Figure 5 molecules-27-08476-f005:**
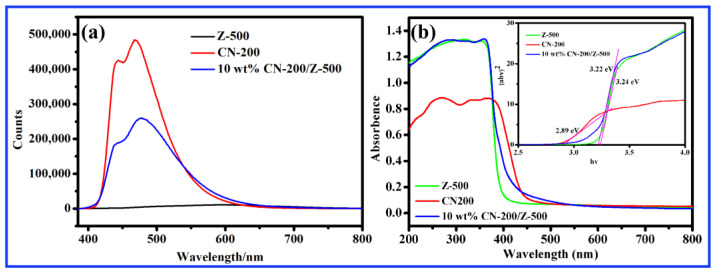
(**a**) Photoluminescence (PL) emission spectra of 10 wt.% CN-200/Z-500 composites, CN-200 and Z-500; (**b**) UV-Visible spectra of 10 wt.% CN-200/Z-500 composites, CN-200 and Z-500.

**Figure 6 molecules-27-08476-f006:**
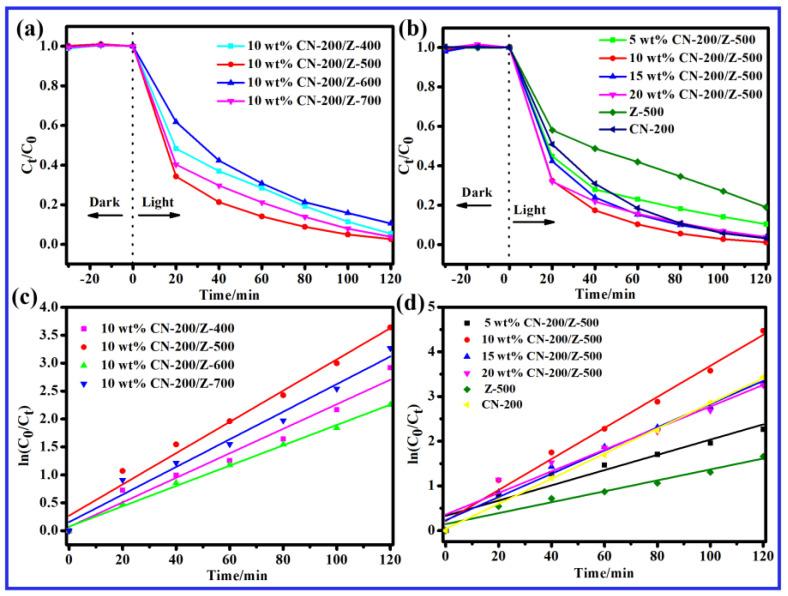
(**a**,**b**) Photocatalytic activity of MB degradation photocatalyst under visible light irradiation; (**c**,**d**) first-order kinetic curve of MB degradation photocatalyst under visible light irradiation.

**Figure 7 molecules-27-08476-f007:**
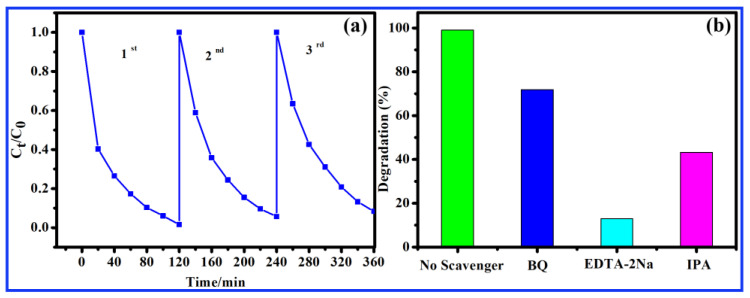
(**a**) Cycle experiments for degradation of MB over 10 wt.% CN-200/Z-500; (**b**) Influence of different scavengers on the degradation of MB over the 10 wt.% CN-200/Z-500.

**Figure 8 molecules-27-08476-f008:**
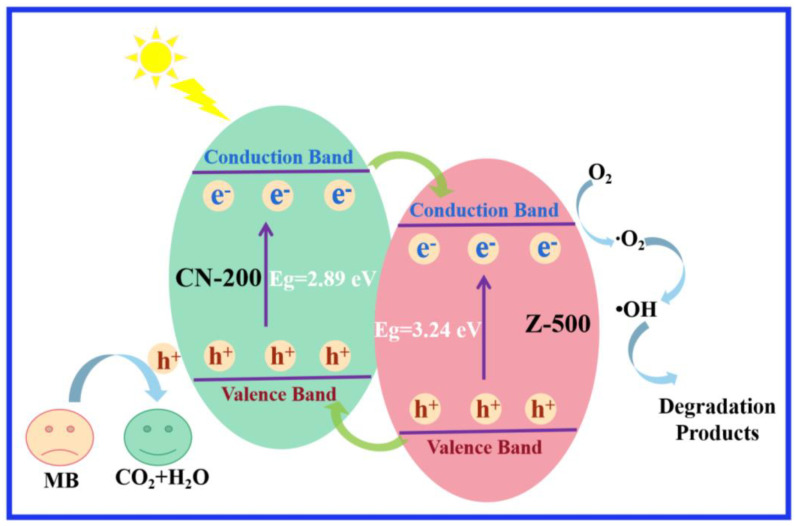
Schematic diagram for the proposed photocatalytic reaction mechanism of 10 wt.% CN-200/Z-500.

**Table 1 molecules-27-08476-t001:** Degradation rate of MB after compound of zinc oxide and CN prepared at different annealing temperatures.

	y = ln(C_0_/C_t_)	R^2^	Degradation Rate (%)
10% CN-200/Z-400	Y = 0.02194X + 0.07045	0.96537	94.6%
10% CN-200/Z-500	Y = 0.02796X + 0.27089	0.97759	97.376%
10% CN-200/Z-600	Y = 0.01819X + 0.07533	0.99565	89.529%
10% CN-200/Z-700	Y = 0.0247X + 0.15485	0.97284	96.195%

**Table 2 molecules-27-08476-t002:** Degradation rates of MB with different catalysts.

	y = ln(C_0_/C_t_)	R^2^	Degradation Rate (%)
5% CN-200/Z-500	Y = 0.01705X + 0.33072	0.92233	89.637%
10% CN-200/Z-500	Y = 0.03473X + 0.21363	0.98704	98.857%
15% CN-200/Z-500	Y = 0.02599X + 0.23163	0.98333	96.249%
20% CN-200/Z-500	Y = 0.02417X + 0.35889	0.95197	96.133%
Z-500	Y = 0.01223X + 0.14589	0.95744	80.971%
CN-200	Y = 0.02809X + 0.03753	0.99826	96.776%

## Data Availability

Not applicable.

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
