# Peer review of "Enhance ZnO Photocatalytic Performance via Radiation Modified g-C_3_N_4"

_molecules, 2022, doi:10.3390/molecules27238476_

Round 1
Reviewer 1 Report
This work focuses on the effect of radiation on composite materials, which is very innovative. In this paper, the effect of radiation absorption dose on photocatalytic performance is fully investigated. The topic is important and the manuscript is well-prepared. Therefore, I would recommend the acceptance of this paper for the publication. There are still some technique issues required further clarification.
1. In the abstract, the abbreviation "CN-200/Z-500" appears directly. I suggest adding the full name.
2. In this paper, there are a few issues of format inconsistency. For example, line 18 has an extra space between wt and %; ZnO:Sb/g-C3N4 in line 54 and ZnO: Sb/g-C3N4 in line 57 have different formats. I suggest that the author carefully check and revise the formatting issues in the full text.
3. DRS test exists in the representation part of the paper, but didn’t mentioned in the discussion section, please add some statements on the DRS test.
4. The individual names in Figure 1 are not formatted in a uniform way, I suggest that they be unified to make the picture more beautiful.
Author Response
Reviewer 1:
This work focuses on the effect of radiation on composite materials, which is very innovative. In this paper, the effect of radiation absorption dose on photocatalytic performance is fully investigated. The topic is important and the manuscript is well-prepared. Therefore, I would recommend the acceptance of this paper for the publication. There are still some technique issues required further clarification.
1 In the abstract, the abbreviation "CN-200/Z-500" appears directly. I suggest adding the full name.
√ Response: This part had been modified and shown in red in the revised manuscript.
2 In this paper, there are a few issues of format inconsistency. For example, line 18 has an extra space between wt and %; ZnO:Sb/g-C3N4 in line 54 and ZnO: Sb/g-C3N4 in line 57 have different formats. I suggest that the author carefully check and revise the formatting issues in the full text.
√ Response: This part had been modified and shown in red in the revised manuscript.
3 DRS test exists in the representation part of the paper, but didn’t mentioned in the discussion section, please add some statements on the DRS test.
√ Response: This part had been modified and shown in red in the revised manuscript.
4 The individual names in Figure 1 are not formatted in a uniform way, I suggest that they be unified to make the picture more beautiful.
√ Response: This part had been modified and shown in red in the revised manuscript.
Reviewer 2 Report
The submitted manuscript deals with construction of g-C3N4 and ZnO composite for photocatalytic application. Graphitic carbon nitride was modified by electron beam. The composite was used for the decomposition of methylene blue (MB). The manuscript cannot be published in its present form and new experiments are necessary. Some comments are given below.
Comments:
1 1. The band-gap energy of ZnO should be mentioned in Introduction.
2 2. There are many papers about the g-C3N4/ZnO composites. Please, explain novelty of this study.
3 3. The description of photocatalytic experiments in Part 4 is very poor.
4 4. DRS and PL spectra are completely missing in Part 5
5 5. Band-gaps were not determined at all.
6. The diffraction (100) of g-C3N4 is missing as well.
6 7. The heterojunction of g-C3N4 and ZnO is not discussed.
8.The photocatalytic decomposition of only one dye (MB) is not enough. See the problems with dyes in literature.
8 9. The constant “k” in the first order rate model is not the Boltzmann constant which is 1.381 10-23 J/K. The rate constant is typical for every reaction and can be determined from experimental data.
9 10. What is the photocatalytic mechanism of the MB decomposition?
1 11. It is not clear how was the g-C3N4 material modified by the electron beam. Please, explain.
Reviewer 3 Report
The authors submitted an article entitled “Radiation construction of g-C3N4 modified ZnO to enhance
photocatalytic performance”. I recommend considering publication after a major revision. Here are my comments:
1. The title is so general and should be revised to a more specific one.
2. Be consistent in using abbreviations. For instance, the “MB” was introduced in line 58, while agin methylene blue was used several times after that.
3. The experiments should be repeated for at least 3 times and the error bars should be added for all the presented data in Figure 5.
4. The recycling of the photocatalyst should be studied to increase the impact of the material.
5. The degradation pathway of the MB should be added.
6. The photodegradation mechanism and electron transformation in the photocatalyst should be studied and a schematic illustration provided by the authors. You can get the idea from the following reference: 10.1016/j.jhazmat.2019.06.018
Round 2
Reviewer 2 Report
The manuscript was revised and can be published now.
Reviewer 3 Report
The manuscript is amended.